# Exploring Egalitarianism: A Conceptual and Methodological Review of Egalitarianism and Impacts on Positive Intergroup Relations

**DOI:** 10.3390/bs14090842

**Published:** 2024-09-19

**Authors:** Rachael J. Waldrop, Meg A. Warren

**Affiliations:** 1Department of Psychology, Western Washington University, Bellingham, WA 98225, USA; 2Department of Management, Western Washington University, Bellingham, WA 98225, USA

**Keywords:** egalitarian, intergroup relations, literature review

## Abstract

Considerable research on intergroup relations emphasizes attitudes, motivations, and emotions that lead to the reduction of prejudice. While factors that actively promote positive intergroup interactions have been emerging, a central theoretical framework has not been formally proposed. To address this gap, we engaged a positive psychology lens to explore how researchers have defined key indicators and consider the positive counterparts of traditional prejudice-reduction models to begin building a new framework of egalitarianism. After scanning 16,840 records that emerged in PsycInfo using keywords “egalitarian”, “non-prejudice”, and “intergroup”, we assessed 158 articles for eligibility. Among the relevant articles (N = 54), we analyzed authors’ definitions, methods of measurement, types of processes, and outcomes associated with egalitarian values as they related to intergroup behavior. Overall, there was notable variability in how researchers conceptualized and studied egalitarianism. We discuss the five broad categories of egalitarianism (prejudice reduction, universal orientation, concern for others, positive expression, and low social dominance orientation) and how they relate to positive or negative and approach or avoidant outcomes. Through these findings, we urge scholars to utilize a centralized model for studying egalitarianism in intergroup contexts moving forward.

## 1. Introduction

Prejudice, bias, stigma, and discrimination are among the most well-studied social psychological phenomena because they are at the root of much intergroup conflict—from microaggressions and individual mistreatment to institutionalized discrimination and wars. Such bias takes root within the individual, and its effects can be observed within socially created systems, both historical and present, and has stimulated robust academic and public research over time. Most recently, intergroup relations in the United States show a complex trend. Research on prejudice and bias has made its way into public discourse and seemed to generate social norms that motivate avoidance of being or appearing prejudiced in pursuit of a more egalitarian society [1,2]. For example, Project Implicit popularized the Implicit Association Task and allowed millions to become familiar with implicit bias [3]. Despite this progress, public views began to shift in 2015, according to data from the Pew Research Center [4]. Now, a majority of Americans report that race relations are poor (58%) and that overt forms of discrimination are more accepted than they used to be (65%) [5]; more women than men (58%) report that being a woman makes it more difficult to be successful in the workplace [6], and LGBTQ-identifying youth report high rates of harassment at school (53%) and negative impacts on their mental health due to anti-LGBTQ policies [7].

Despite the breadth of awareness, resources, and national attention paid to bias and prejudice over the past 25 years, the state of intergroup relations continues to be concerning. The relationship between people’s internal values and external social norms and that relationship’s influence on behavior still plagues and intrigues social psychological researchers. For instance, even though over 75% of working adults report intentions to be allies, a majority hesitate to act [8]. Yet some do speak up against injustice and actively build high-quality supportive relationships with marginalized and stigmatized individuals. What differentiates those who step up as allies, despite the fear and discomfort, from those who remain silent and withdrawn?

In pursuit of this answer, social psychologists have explored the extent to which internal processes motivate sincere behaviors, like seeking to reduce harm to others, compared to more surface-level behaviors like avoiding saying the wrong thing and being expunged from social platforms (e.g., “being cancelled”). Since Plant and Devine’s [9] foundational research on internal and external motivations to be egalitarian, numerous studies have examined a variety of outcomes resulting from egalitarian values. Indeed, motivators like goals, values, and internalized beliefs have been found to influence positive behavior change, yet much of this research has heavily focused on prejudice avoidance (i.e., non-prejudice), stereotype reduction, and bias neutralization, despite referencing “egalitarianism”. There has been a lack of clarity on how egalitarian processes may be distinct from prejudice avoidance processes.

We believe positively framed processes, like the pursuit or valuing of egalitarianism, may be a rich avenue of exploration to understand how true allies cultivate authentic intergroup relationships and experience positive intergroup dynamics, above and beyond eradicating prejudice. Importantly, though, there seems to be no consensus on a unified definition of egalitarianism. Research that directly studies egalitarianism as distinct from non-prejudice has been sparse and sporadic. To date, there has also been no synthesis that consolidates the various and disconnected streams of research on the conceptual parameters of egalitarianism, egalitarian motivations, behaviors, or the conditions under which it engenders positive social change. In the absence of such an integration, the field not only risks duplication of efforts but also limits the development of theoretically rich advancements and practical interventions for fostering positive intergroup relations. To address this gap, the current paper critically reviews these various conceptualizations and operationalizations of egalitarianism in support of building a cohesive framework of egalitarianism.

The distinction between egalitarianism and prejudice avoidance as concepts is important because a positive-oriented concept such as egalitarianism has the potential to influence more pro-social and positive intergroup behaviors, above and beyond avoidance-oriented behaviors that are driven by prejudice-avoidance processes. Even well-meaning allies can successfully avoid behaving in prejudiced ways, but that does not always translate into proactive behaviors of intergroup support [10]. Therefore, the purpose of the current paper is to synthesize the research that has already developed around egalitarianism, highlight theoretical concerns particularly around conflations with non-prejudice, critically analyze methodological gaps, and offer an agenda for future research that can move the field forward.

### 1.1. Theoretical and Methodological Contributions

Egalitarianism has been referenced throughout social and psychological scholarship, varying in application from economic and political theory to philosophical and psychological inquiry [11,12,13]. While even recent scientific developments reference egalitarianism in characterizing motivations, outcomes, and goals, there exists no unifying framework of egalitarianism that makes sense of these various conceptualizations. To help advance this line of research, we map the various ways in which egalitarianism has been operationalized in psychological research to identify the underlying dimensions of egalitarianism as it has been studied. We focus on the discipline of psychology to tie the concept to cognitive and motivational processes that ultimately influence social behaviors. This conceptual review allows us to identify potential gaps in theorizing and propose a theoretical focus for research moving forward.

In addition to mapping the conceptual landscape of egalitarianism, we consider the ways it has been measured and methodologically developed over time. For instance, by examining the types of measurement scales used in self-report studies, we demonstrate the various assumptions being applied to the concept. By considering the types of experimental manipulations and paradigms used, we assess the (un)predictability of outcomes. This synthesis allows us to consolidate the various ways egalitarianism has been measured and offers scholars a useful resource that points to variances, gaps, and opportunities for future research development.

Finally, we examine the reported outcomes associated with egalitarianism to map the types of behaviors (i.e., avoidant or approach-oriented) with which the concept is associated. Through this, we reveal how definitions and methods themselves render either positive or negative outcomes, reinforcing a need for a centralized conceptualization in the field. For example, associating egalitarianism with “creating equity and belonging” may stimulate different outcomes than associating it with avoiding discrimination. Highlighting the variance in outcomes as a direct result of theoretical and methodological considerations serves as a call for researchers to align investigations moving forward.

We determined that using a review method to map the landscape of egalitarianism conceptualization and research was appropriate for our purpose, since our goal was to analyze how the concept of egalitarianism itself has been (inconsistently) defined, operationalized, measured, and studied. Further, our critical analyses aimed to identify the gaps and point to how the study of egalitarianism could be advanced if it included an ‘approach orientation’ in its definition, operationalization, and measurement. This would also open the field to examine a new and different set of potential outcomes, above and beyond reductionistic outcomes.

### 1.2. Present Study

In the present paper, we sought to answer how researchers have defined, operationalized, and measured (e.g., self-report, manipulation, etc.) egalitarianism and how, as a result, intergroup outcomes have been impacted. Our research questions were as follows:(1)How has egalitarianism been defined in the social psychological literature?(2)How has egalitarianism been measured across the social psychological literature?(3)How does the variability in definition and design influence the outcomes rendered?

Definitions of egalitarianism were coded based on authors’ descriptions in the introductions and literature reviews. When a definition was absent from the introduction, measurement indicators were used. Using a bottom-up coding technique, we organized outcomes by their positive/negative and approach/avoidance implications. First, we determined whether results led to positive outcomes for intergroup relations (e.g., more positive outgroup attitudes, less stereotyping, more positive intergroup behavior, etc.) or negative outcomes (e.g., less positive intergroup behavior, more outgroup stereotyping). Second, we determined whether the results were associated with the absence of unwanted outcomes (e.g., the absence of bias, the inhibition of stereotyping, the absence of discriminatory behavior) (coded as avoidant) or with the presence of desirable outcomes (e.g., positive attitudes, pro-social helping behaviors, positive nonverbals) (coded as approach), or a mix of both. Lastly, we noted the various moderating factors present in each study.

## 2. Materials and Methods

We searched for peer-reviewed articles published before May 2024 within the PsychInfo electronic database using keywords “Non-prejudiced” OR “Egalitarian” AND “Intergroup”. Due to our focus on the relationship between these keywords, we narrowed the search by simultaneously searching “non-prejudiced” and “intergroup” first, then “egalitarian” and “intergroup” second. To supplement this, we conducted additional searches within the following journals to ensure that articles in developmental, counseling, and ethnic minority disciplines were not missed: Developmental Psychology, Applied Psychology, Community Psychology, Counseling and Clinical Psychology, Counseling Psychology Quarterly, Counseling Psychology, and Cultural Diversity and Ethnic Minority Psychology. These additional searches yielded two unique results. Finally, we used keywords “control prejudice” or “without prejudice” with “egalitarian” to capture any additional studies. This search yielded 15 unique results. In total, searches with these unique combinations yielded a total of 16,840 results.

Articles were included for analysis if the following three criteria were met: (1) egalitarianism or non-prejudice were included as measured or manipulated predictor variables, (2) intergroup paradigms reflected existing social identities based on real-world dynamics (e.g., gender, race, socioeconomic status, nationality, sexual orientation, etc.), and (3) articles were published in English. We screened a total of 158 records based on the inclusion and exclusion criteria.

## 3. Results

A final sample of 54 articles were rendered as eligible for the current analysis. Using a bottom-up coding technique to categorize authors’ definitions of egalitarianism, five unique domains emerged: prejudice avoidance, universal orientation, concern for others, positive expression, and low social dominance orientation. Authors describing “prejudice-avoidance” egalitarianism used the term synonymously with “non-prejudice” and “absence of bias”. Definitions of egalitarianism as “equality” or “equal treatment for all” were coded as “universal orientation”. “Concern for others” egalitarianism was described by authors as genuine concern for others’ well-being, beyond possessing the generic value of equal treatment. We coded authors’ descriptions of egalitarianism in reference to positive behaviors toward outgroup others as “positive expression” egalitarianism. Finally, “low social dominance” egalitarianism was categorized based on a growing body of research that uses the disapproval of hierarchical systems of oppression as a representation of egalitarian beliefs.

We coded the methodological processes used across all studies (e.g., self-report, manipulation, observational, or scale development). In addition, we coded whether the outcomes in each study were related to an increase in desirable behaviors (approach-orientation) or a decrease in undesirable behaviors (avoidant-orientation). To see the breakdown of outcomes by each coded definition, see Appendix A. When present, we also highlighted the various moderators that influenced the expression of egalitarianism across contexts, accounting for individual and environmental differences.

Next, we present five ways in which egalitarianism has been conceptualized in more detail, namely, prejudice avoidance, universal orientation, concern for others, positive expressions towards outgroups, and low social dominance orientation. For each, we provide the definition and operationalization of the concept as purported by authors, methods used to study it, the common outcomes measured, moderators affecting them, and our critical analysis of the empirical scholarship.

### 3.1. Prejudice Avoidance

Egalitarianism was most commonly conceptualized as the absence of prejudice, bias, or discrimination such that egalitarianism is situated at the polar end of a singular dimension reflecting prejudice. Across 20 articles, participants who displayed low or null expressions of bias were considered egalitarians. Methods of measurement included self-report scales, implicit association assessments, or attitude assessments of specified outgroups. Most outcomes were positive yet avoidant-based, meaning that egalitarianism was associated with lower degrees of bias or discrimination. Across all studies in this category, there were over twice as many avoidant-based indicators (20) compared to approach-based indicators (12). About half of the studies in this domain identified moderators that were essential in attaining a positive outcome, suggesting that additional conditions influenced the expression of egalitarianism. A list of articles, method type, and outcomes for prejudice avoidance are shown in Table 1 below.

#### 3.1.1. Methods

Prejudice avoidance was most commonly measured via self-reports of bias. A widely used measure of prejudice avoidance was Plant and Devine’s (1998) Motivations to Respond without Prejudice scales. These scales distinguish between internal motivation reflecting personally important values that guide non-prejudiced behavior and external motivation reflecting social pressures for complying with non-prejudiced social standards [9]. Other common scales used to represent egalitarianism were Brigham’s (1993) Attitudes Toward Blacks (ATB) scale [15,19], the Ring measure of Social Values [22], the Motivation to be Non-prejudiced scale (MNPS) [27], and the Scale of Prejudice and Egalitarian Goals [30]. Only one study used an open-ended response format to capture organic definitions of egalitarianism among Black respondents by asking “What are some things that you have personally seen or heard a white person do or say that made you think that person was not prejudiced?” and “In general, what can white people do to let you know they are not prejudiced, if they really are not?” [25].

Five studies used implicit bias measures, typically measured by reaction times to associated stimuli or stereotype identification tasks (e.g., the Implicit Association Task, Lexical Decision Tasks, Weapons Identification tasks). In these tasks, the time it takes participants to correctly identify ingroup and outgroup stimuli, associate positive stimuli with ingroup or outgroup members, or determine whether or not to virtually “shoot” an ingroup or outgroup target are all compared. When there was a shorter or null difference in reaction times between ingroup and outgroup target stimuli, participants were considered to not possess implicit bias and therefore encompass egalitarianism.

Eight studies utilized manipulation techniques to elicit egalitarian ideals through social norm priming, which involved making salient others’ disapproval of discrimination towards outgroup members. Primes were delivered overtly, through direct (false) feedback about privileged group members’ disapproval of social policies that favored other privileged groups or through videos of others confronting prejudice, and subtly, such as words on a t-shirt worn by an experimenter, an article about inequality, or simply being in the room with other people (i.e., who presumably would make salient prominent social norms).

Finally, one study measured egalitarian goals by providing participants with false feedback about their stereotypical behaviors and measuring their compensatory behaviors on subsequent stereotype activation tasks [14]. In this study, stereotype suppression following a perceived violation of egalitarian standards represented successful prejudice avoidance indicative of egalitarianism.

#### 3.1.2. Outcomes

Across self-reported and manipulated methods, egalitarianism construed as prejudice avoidance was almost unanimously predictive of positive results, yet there were almost twice as many avoidant measures used (20) compared to approach measures (12).

Prejudice avoidance was positively related to self-reported positive attitudes toward outgroup members and inversely related to negative attitudes such as modern racism, authoritarianism, and protestant work ethic beliefs. Emotionally, participants across these studies tended to report focusing more on positive effects when reflecting on their motivations and outgroup imaginary interactions and more negative emotions when reacting to anti-outgroup primes. Behaviorally, studies measured and showed participants tend to favor outgroup members in experimental tasks (e.g., point-allocation tasks) and spend more time adjusting their behavior to decrease bias. Overall, outcome metrics were overwhelmingly self-reported, limiting inferences about behavioral impacts. Further, although outcomes were positive, they were overwhelmingly avoidant-framed, revealing that the impact on intergroup relations is limited to harm reduction.

Studies utilizing manipulation methods wrought mixed results. Overall, manipulated forms of prejudice-avoidance egalitarianism led to a successful reduction in undesirable attitudes and behaviors but did not necessarily inspire positive, proactive intergroup dynamics. One study that measured perceptions of egalitarian confrontations found more positive associations of egalitarian behavior than non-egalitarian behaviors. However, this study did not include a behavioral outcomes measure. One study found negative effects such that receiving positive feedback about one’s egalitarian values displayed more biased behavior in supplemental tasks. Overwhelmingly, the outcome measures were still avoidant-framed. Furthermore, given that a manipulation study revealed an increase in displays of bias after exposure to an egalitarian prime exemplifies the disconnect between self-reported outcomes and actual outcomes, resulting in the need for more observed or behavioral outcomes beyond self-report measures.

#### 3.1.3. Moderators

Across the aforementioned studies, four types of moderators influenced the outcomes associated with prejudice-avoidant egalitarianism: motivational source, group dynamics, individual characteristics, and culture.

Internal motivation led to positive attitudes and successful stereotype inhibition, whereas external motivation led to negative outgroup attitudes and increased stereotyping, bias, and discrimination [9,26]. Because internal and external motivation are separate dimensions, oftentimes, the presence or absence of external motivation can override the positive effects associated with prejudice avoidance. This indicates that although internal motivation is related to positive outcomes, some stimuli may not bring this motivation into awareness and subsequently fail to inspire positive behaviors.

Group dynamics largely influenced the effect of egalitarianism on intergroup outcomes. Although social norming curbed prejudice, moral licensing and group distinctiveness threat negated this effect. Moral licensing, the act of using prior positive behavior to justify subsequent anti-social behavior, increased rather than decreased prejudice towards outgroup members. Inducing similarities between ingroups and outgroups (termed group distinctiveness threat) also led to increased prejudice and stereotyping, despite the presence of an egalitarian social norm prime [31]. When group similarities were manipulated to feel small, egalitarian social norm primes did reduce prejudice, but when similarities between groups were emphasized, egalitarian norms did not reduce participants’ prejudice against outgroup members. Taken together, these studies reveal that drawing awareness to egalitarian social norms, even among fellow ingroup members, does not always reduce prejudice.

There were two individual difference characteristics identified that influenced the effectiveness of egalitarianism: Optimism and gender. These moderators were observed in the context of behavioral outcomes of bias confronting behavior and a stereotype-placement decision task, relaying the importance of individual characteristics in predicting actual intergroup behavior, beyond attitudes and self-reported emotion states.

Finally, cultural background or setting significantly influenced implicit attitudes and behaviors. In particular, organizations and industries made up predominantly of female employees tended to display less stereotyping behavior. Self-reported collectivists (as compared to individualists) tended to be more positively impacted by egalitarian primes in terms of their attitudes and bias scores. Thus, cultural context and environmental setting may influence the degree to which social norms produce egalitarian-favoring outcomes.

#### 3.1.4. Critical Summary

Conceptualized as the desire to control, reduce, or rid the self of internal bias, prejudice avoidance was the most common conceptualization of egalitarianism within intergroup research, according to authors’ purported definitions. In this way, egalitarianism was interchangeable with non-prejudice rather than studied as a separate construct. A variety of methodologies ranging from scale development and self-report surveys to indirect cognitive tasks and experimental manipulations converged to portray the generally positive effects of this construct. However, a majority of these positive outcomes were measured via avoidant-framed instruments such as low scores of prejudice and bias, lack of negative implicit associations, and the suppression of stereotype activation. Simply put, findings conveyed that people who sought to avoid or suppress bias were more likely to avoid or suppress bias. In addition, various moderators either weakened or shifted the positive effects. Specifically, external motivation, group distinctiveness threat, moral licensing, gender, and cultural context shifted the effects of prejudice avoidance from positive to negative.

Despite the effect that this form of egalitarianism had on the reduction in negative intergroup processes, we found some limitations with this conceptual approach. First, prejudice avoidance was largely observed and measured in relation to intrapersonal consequences (i.e., cognitive biases, emotional outcomes, self-concepts), but few studies observed how these prejudice-avoidant mechanisms influenced outgroup others, especially in situations that would be found outside a laboratory setting. Intergroup relationships naturally involve interactions that contain differing perspectives and experiences. As such, interpersonal experiences are imperative to consider when assessing the impact of intergroup motivations. Thus, researchers might misrepresent the state of intergroup attitudes and behaviors to convey a society becoming less biased and more harmonious, when the research only shows reductions in bias within the self, not necessarily an increase in pro-social, outgroup harmony.

Second, these studies were based largely on self-reported data. Predictions on how these self-reports translate to actual behavior are limited without additional studies to corroborate. Lastly, equating egalitarianism with the absence of or reduction in prejudice stifles opportunities to explore positive, proactive outcomes like intergroup flourishing. Prejudice reduction and regulatory processes could be a key component of egalitarianism but should not encompass the entire concept. Utilizing a positive psychological approach and centering intergroup flourishing is thus encouraged in re-theorizing egalitarianism as a guiding value and source of motivation.

### 3.2. Universal Orientation

The second most common conceptualization of egalitarianism reflected a general support for the belief that all people deserve to be treated equally. From this theoretical orientation, 14 articles treated egalitarianism and prejudice as co-existing dimensions whereby individuals could possess high or low levels of both simultaneously. Common measurement methods ranged from self-report surveys to behavioral observation and experimental manipulations. Universal orientation led to a mix of harmful (6) and desirable outcomes (9) that were overall more avoidant-based (8) than approach-based (5). A list of articles, method type, and outcomes for universal orientation are shown in Table 2 below.

#### 3.2.1. Methods

Universal orientation was investigated through self-report measures (8), manipulations (5), and observational (1) methods. Many self-reported measures were either scale developments, open-ended response options, or broader measures capturing value identification and attitudes of outgroup others. These scales varied broadly in what they captured: role bias, perceived similarities between the self and others, beliefs in equal access to resources and rugged individualism, and perceptions of egalitarian social norms. Manipulation tasks used to induce egalitarian social norms, identities, or attitudes consisted of mismatching traditional roles like describing men who are feminist or using written cues promoting ideals of equality such as “everyone is the same and therefore should be treated equally”. Even in an intervention setting, egalitarianism was measured based on how youth participants responded to traditional or sexist gender roles.

#### 3.2.2. Outcomes

Though majority of the studies revealed that egalitarianism led to desirable outcomes (9), many studies also showed harmful consequences associated with universal orientation (6). Unlike prejudice avoidance, there was much more variability in the extent to which universal orientation was regarded as a pro-social or anti-social process. Positively, egalitarianism was related to positive attitudes toward outgroup others, low levels of prejudice, bias, and outgroup anxiety, self-reported volunteering behaviors, and feelings of being connected to others.

Among the positive outcomes were the increase of pro-social processes (approach) and decrease of undesirable processes (avoidance) whereby manipulation methods were more often measured by avoidant-framed outcomes like decreases in bias and discrimination. In one case, highly prejudiced participants who wrote about treating people equally were no more likely to identify avoidance words than approach words following the target prime, indicating that even though egalitarianism and prejudice may be distinct cognitive processes, egalitarianism may still function to curb the effects of prejudice. Negatively, egalitarianism was found to be correlated with hostile attitudes and behaviors.

In the limited studies where self-reported universal orientation was measured against behavior, egalitarianism was associated with an increase in bias awareness but was not found to later impact stereotyping behavior. Negative results were due to moderators that were identified as variability in equality attainment, demographic and identity characteristics, low group distinctiveness, and environmental ambiguity/situational justification.

Many of the outcome metrics were self-reported measures, again posing limitations on how actual behavior or intergroup relationships would be impacted. In one unique study, Black adults rated their perceptions of how egalitarian or prejudiced White responses were to a written prompt [50]. Based on their writing alone, Black participants were able to accurately predict levels of racism and internal motivations to be egalitarian [9] based on their humanizing language and support for equality of opportunity. White participants who wrote about the personal responsibility of Black people and the current existence of equal opportunity were seen as more prejudiced and less sincere. This highlighted that although White individuals have an array of beliefs about egalitarianism, Black adults are attuned to cues that reflect subtle levels of prejudice. This study did not include behavioral measures but did reflect the importance of cross-group interaction in studies whereby the impacts of egalitarianism can be measured beyond self-reports.

#### 3.2.3. Moderators

Considering individual differences, there were some groups for which appeals to this belief consistently produce harmful intergroup consequences. For example, individuals who subscribe to meritocracy or rugged individualism may have reported holding egalitarian values yet displayed a negative outgroup bias. This highlights that within this concept, there are more positive and potentially more harmful conceptualizations that individuals hold when confronted with the idea of equality or at least equal social attainment. Thus, beliefs about how equality is attained may differ between those who endorse the Protestant work ethic and those who endorse humanitarianism, so individual differences are important in capturing this nuance.

Similar to prejudice avoidance, three studies found that women tended to show more egalitarian attitudes than men, and men or masculine-related characteristics were related to more negative outcomes. Specifically, women tended to have more favorable outgroup attitudes and more favorable attitudes toward egalitarian (or feminist-identifying) men and felt less distinctively different from men. Certain aspects of the male social identity were revealed to be less associated with egalitarianism and more associated with harmful behaviors (e.g., sexism and propensity to sexually harass). Relatedly, manipulations that reinforced similarities between social groups (i.e., low group distinctiveness) tended to activate negative outgroup attitudes among men but not women, even following egalitarian primes [49]. Thus, men tended to engage in less egalitarian processing and more discriminatory behaviors, even following egalitarian primes.

Emphasizing the importance of the environment on influencing behavior, a string of studies in this domain revealed overwhelming support that the explicitness of social norms dictated group behavior. When norms were clear, less ingroup favorability was observed compared to when norms were absent or ambiguous. Based on a robust number of studies [52], these findings suggest that egalitarian norms only influenced behavior and attitudes if they were clear and explicit. Therefore, universal orientations of egalitarianism may vary in the extent to which these measures capture internal versus external motivations.

#### 3.2.4. Critical Summary

Universal orientation was conceptualized as the general philosophy that all people are equal and deserve equal rights. Consequently, egalitarianism was construed as a process operating independent from prejudice. Whether universal orientation led to desired or undesired outcomes varied considerably based on the use of self-reported or manipulated methods. Self-report measures typically were associated with perceptions of similarity between the self and diverse others, more positive outgroup attitudes, less negative outgroup attitudes, and less discrimination. These positive outcomes were stronger when overt and direct egalitarian social cues were present in the environment. In their absence, people typically displayed an ingroup bias.

Whereas some primes led to decreased cognitive avoidance, increased awareness of sexism among children, and positive attitudes and behaviors towards outgroup members, overwhelmingly, egalitarian primes tended to prompt outgroup bias, aggressive behaviors, and increased stereotyping. Some key factors that exacerbated these negative effects were underlying interpretations of egalitarianism (i.e., Protestant work ethic labeled as egalitarianism), masculinity, group distinctiveness threat, and ambiguous social norms (i.e., the lack of clear egalitarian norms). This discrepancy reveals there are likely differences between how a person explicitly reports their egalitarian values and how they react or engage when responding to implicit cues. These reveal the importance of moving beyond self-report measures so researchers can uncover more natural and authentic displays of egalitarianism.

Based on the variability in outcomes that was directly related to the measurement type (i.e., self-report versus manipulation), there are likely underlying processes being conflated with universal orientation egalitarianism or individual differences that should be accounted for. The inconsistency in these findings could be due to variations within personal endorsements of egalitarianism such that people who hold strongly individualistic, identity-blind (e.g., color-blind) beliefs possess high levels of prejudice that are only detectable through behavioral measures. Self-report studies that did not utilize a variety of outcome measures, particularly behavioral outcomes, or that did not measure differing beliefs about the attainment of equality would fail to detect these differences. Moreover, self-report measures related to social beliefs could be particularly prone to social desirability bias, resulting in the observed discrepancy. Without specific definitions and theoretical parameters guiding this research agenda, many of these observed results may be reflecting a multitude of different processes like the Protestant work ethic, humanitarianism, or external motivation only, which would sporadically result in some positive and some negative intergroup outcomes.

The lack of nuance within universal orientation egalitarianism, in addition to an over-reliance on self-report methods, has likely led to the unclear pattern of findings in the literature. Although people generally strive to hold egalitarian beliefs that promote equality among all social groups, co-occurring beliefs related to prejudice, essentialism, or system justification may influence and subvert the potentially positive effects of egalitarianism. Hence, a more nuanced measure and a robust combination of measures may be needed to gain insight into egalitarian processes. Acknowledging the existence of structural racism, current conceptions of egalitarianism should encompass existing, covert social barriers and the pursuit of fairness for marginalized groups who continue to lack equal power and status in society [53]. In fact, emerging work in counseling psychology has focused on attitudes, norms, and behavioral intentions related to social justice and advocacy to be inclusive toward a growing multicultural population. Additionally, work addressing the practice of allyship and its underlying motivations has become increasingly popular [54,55,56]. An avenue for future exploration would be to study the beliefs and values among those who engage in allyship and advocacy.

### 3.3. Concern for Others

A small section of the literature conceptualized egalitarianism as humanitarianism or a sincere concern for others and social justice. Similar to universal orientation, this construct was conceptually distinct from prejudice, but unlike universal orientation, this concept focused more on people and their experiences rather than the generalized belief of equality. Four studies fell under this domain, relying solely on self-report methods and showing exclusively positive results. The exclusive reliance on self-reports limits the ability to generalize these findings to behaviors across time and situations and, ultimately, its role in influencing positive intergroup relations. A list of articles, method type, and outcomes for concern for others are shown in Table 3 below.

#### 3.3.1. Methods

The Humanitarian-Egalitarianism (HE) scale was developed by Katz and Hass [41] and captures self-reported values that all people, especially disadvantaged groups, should be cared for by society. This scale was subsequently used to measure attitude correlates of ingroup and outgroup others and online social interaction with outgroup members. One study measured the impact of egalitarianism on identity development and well-being among multiracial individuals using a self-report measure of caretaker egalitarian socialization.

#### 3.3.2. Outcomes

Concern for others was related to exclusively positive outcomes that were predominantly approach-oriented, such as possessing pro-Black attitudes, and strengthened perceptions of inequality, which promoted support for social action and policies that served disadvantaged groups, and the tendency to have more diverse social networks. Studies also showed concern for others to be negatively associated with bias and prejudice. Among communities of color, egalitarian socialization was associated with positive racial attitudes, positive upbringing, and positive moral influences. Examining egalitarianism and how it has shaped identity and development among people of color was unique to other streams of egalitarian research, which have primarily explored egalitarianism’s impact on traditionally majoritized groups (e.g., White, male, heterosexual) and how it impacts others. This study found positive effects on the well-being, developmental trajectory, and socialization of individuals of color—a small representation in the literature comparatively.

The emergent pattern whereby self-report measures were associated with positive results holds true here. Having established that self-reports were related to positive results and that other measures revealed negative results in previous sets of studies, diverse metrics are needed to (dis)confirm how this version of egalitarianism positively impacts intergroup dynamics. Despite this critical limitation, research in this field broadens who has typically been considered as benefiting from egalitarianism and how. Whereas typically socially advantaged members are the primary sample for these studies, more diverse samples have been represented here.

#### 3.3.3. Critical Summary

Concern for others, characterized by an endorsement of humanitarianism and social justice, was associated exclusively with positive intergroup outcomes. Socially privileged group members who were concerned for others had more positive outgroup attitudes, less prejudice, and interacted with outgroup members more frequently on online platforms compared to individuals who were not. Among people with marginalized social identities, sincere concern for others that was modeled by caretakers influenced positive racial identity development, high self-esteem, and general well-being.

The research conducted under this conceptual delineation is limited as only four articles used this characterization and exclusively relied on self-report methods. The intrapersonal mechanisms that influence and are influenced by this construct are thus still unknown. Still, this limited sample of research has offered insight into some nuances that should be considered in the development of a central framework of egalitarianism. First, concern for others was a person-centered belief system, unlike previous conceptualizations that were focused on generic beliefs about equality. Second, outcomes were directly measured among people from socially disadvantaged groups, at least in one qualifying study. Research from the perspective of multiple social groups helps us to better understand the full scope of egalitarianism’s effectiveness and provides a valid, reliable understanding of values that impact intergroup relations.

Concern for others was related to low levels of prejudice and high levels of positive outgroup attitudes, outgroup contact (online), and social justice support, but these relationships were cross-sectional. Thus, it is unknown whether concern for others aids in the processes of actively reducing bias and prejudice (avoidance) or motivates benevolence towards outgroup others (approach). Understanding these relationships is vital for predicting behavior and structuring practices like bias training and social justice education, garnering public support for social policies, and motivating positive intergroup behavior.

### 3.4. Positive Expressions towards Outgroups

Across five studies, a behavioral domain of egalitarianism emerged which consisted of positive expressions toward, favorable ratings of, and intentions to approach outgroup others. This form of egalitarianism, coded as positive expressions towards outgroups, emphasized the interpersonal function and expression of egalitarianism. Two studies relied upon self-report methods, two studies used manipulation methods, and one study was a meta-analysis. Half of the results revealed positive, desirable outcomes that were approach-oriented; however, two identified moderators curtailed these positive effects. Articles and corresponding information are shown in Table 4 below.

#### 3.4.1. Methods

Because egalitarian domains were coded and thematically organized according to authors’ definitions, studies using the internal and external motivation to respond without prejudice scales were included here as the authors defined the construct as promoting positive egalitarian behavior, as distinct from prejudice avoidance (external) [61]. The way this construct was represented reveals (1) the variability in conceptualizations over time and (2) how methods and outcomes themselves have been guiding the interpretation of the concept, rather than theory.

Other studies in this domain utilized interview methodology to capture egalitarian socialization behaviors and manipulation methods to alter perceived states in relation to egalitarian self-concepts. These manipulations led participants to think they were either closer or further away from egalitarian ideals of treating outgroup members as equals.

One meta-analysis was conducted to understand the contextual factors that influenced favorable responses and behaviors toward Black people. Included in the analysis were 31 empirical studies that investigated how egalitarian-promoting environments versus ambiguous environments influenced the evaluation of White versus Black targets. In contrast with Dovidio and Gaertner’s [37,38,43,51,52] line of research on egalitarian social norms, which was discussed previously under universal orientation, Aberson and Ettlin [60] were particularly interested in the processes and correlates of positive egalitarian behaviors and expressions, rather than environmental aspects of social norms.

#### 3.4.2. Outcomes

Self-reports of positive expressions toward outgroup members were related to positive outcomes for both socially privileged and marginalized groups, such as self-reported and observed approach behaviors during imagined and real interracial interactions, pro-outgroup and pro-diversity attitudes, and positive reports of intergroup interactions. When self-concepts were altered via manipulation methods, however, negative outcomes arose such as reductions in authentic and supportive behaviors, high bias scores, and avoidant behavior. Negative outcomes were found both when people were induced with egalitarian congruencies and egalitarian discrepancies, meaning there were consequences when participants felt they were close to (e.g., congruent) their egalitarian identity and when they were distant from (discrepant) their egalitarian identity. Again, here we observe a pattern whereby researcher methods are predictive of the types of outcomes (e.g., positive, negative, mixed), such that self-report methods tend to be more positive than any other methods.

In the meta-analysis, clear egalitarian environments engendered pro-Black responses whereas ambiguous environments engendered pro-White responses [60]. Comparing majority White samples and majority Samples of Color, an ingroup bias emerged within ambiguous environments such that Black participants favored the Black target and White participants favored the White target. Within egalitarian environments, however, the Black target was generally favored regardless of participant race. Thus, White individuals seemed to display positive expressions toward outgroup members when social norms explicitly promoted positive expressions toward others.

#### 3.4.3. Critical Summary

Positive expression was defined by positive or favorable expressions toward outgroup members, functioning as a behavior-based conceptualization of egalitarianism. Self-report measures were related to positive attitudes and intergroup behaviors that were approach-oriented (e.g., smiling and making direct eye contact when speaking with someone from a different race). Manipulated egalitarian self-images, whether boosted or threatened, led to harmful outcomes for people of color, such that individuals exhibited more implicit racial bias and avoidance behaviors, even after receiving positive feedback about their progression on egalitarian goals.

Due to the small number of studies developed under this conceptual framework, limited conclusions about the nature of positive expression can be drawn, beyond the recurring theme that self-report measures tend to result in positive outcomes. The motivational processes responsible for influencing the outcomes found among the self-report studies were unexplored, beyond the possession of an internal desire to eschew prejudice. In conjunction with previous studies that displayed a link between internal motivation to respond without prejudice and the successful inhibition of prejudiced responding, there seemed to be an unexplored underlying factor of internal motivation that drives both the inhibition of prejudice and the promotion of approach behaviors and strategies. Moreover, externally sourced cues, such as egalitarian norms and primes that elicit an egalitarian self-image, seemed to spuriously predict positive and negative outcomes. A continued development of how internalized values are related to external expressions and behaviors, and how the environment influences this connection, would broaden insights into this conceptualization of egalitarianism.

### 3.5. Low Social Dominance Orientation

In the most recent work, egalitarianism was conceptualized in terms of attitudes towards structural equality and social hierarchy. This structure-based concept was predominantly operationalized via low levels of social dominance orientation or disagreement with the belief that social hierarchies exist for a reason and should not or cannot be challenged. Because social dominance beliefs were the central focus of these studies, with high social dominance representing anti-egalitarianism and low social dominance representing egalitarianism, we identified this collection of studies as “low social dominance orientation (SDO)”. Low SDO was observed in 11 articles and was predominantly measured through self-report methods, although one study utilized a manipulation technique. A list of articles, method type, and outcomes for low social dominance are shown in Table 5 below.

#### 3.5.1. Methods

Altogether, eleven articles comprised this domain, with ten studies relying on self-reports of attitudes and beliefs and one study utilizing a manipulation technique. The self-report studies relied on the Social Dominance Orientation scale [60]. This scale consists of subscales measuring pro anti-egalitarianism and con anti-egalitarianism. Anti-egalitarianism was developed as a sub-dimension of social dominance, designed to capture beliefs of people who subtly oppose social change that promotes equality (e.g., pro anti-egalitarianism: “We shouldn’t try to guarantee that every group has the same quality of life” and con anti-egalitarianism: “We should do what we can to equalize conditions for different groups.”). One self-report study relied on cultural-level indicators of hierarchical versus egalitarian societies to explore the relationship between prejudice and intergroup contact. One study asked participants to rate how egalitarian vs. fair their ingroup norms were. Finally, one study utilized manipulation techniques to alter regulatory approaches to responding to structural inequality (e.g., ideal vs. ought) [78].

#### 3.5.2. Outcomes

Overall, low SDO was related to generally positive (10) outcomes that benefitted traditionally disadvantaged or underserved social groups, though outcomes tended to be both approach (7) and avoidant (8). Certain moderators, such as regulatory focus, group memberships, and colorblindness interacted with low SDO to produce negative (8) outcomes. Positive contact between advantaged and disadvantaged groups was positively related to positive outgroup attitudes among disadvantaged group members. Disadvantaged group members who held positive outgroup attitudes tended to perceive the advantaged group as fair and focused less on inequality. In turn, these factors led to disadvantaged group members holding less support for structural social change, an important component of enhancing social status. Though intergroup attitudes may be more positive as a result of positive contact, when disadvantaged group members’ beliefs in inequality are low, they may pay the price.

In the development of the SDO-Egalitarian sub-dimension, egalitarianism was found to be negatively related to political conservatism, system-justifying beliefs, support for unequal resource distribution, opposition to equality-driven policies, and subtle prejudice. Egalitarian beliefs also predicted the tendency to be more aware of societal inequality between social groups, which had the potential to subsequently affect support for social change policies. Assessing the impact of social dominance beliefs, egalitarians and anti-egalitarians alike felt empathy for groups that they perceived experienced harm; however, egalitarians perceived harm and directed support towards disadvantaged groups, whereas anti-egalitarians perceived harm and directed their support toward advantaged group members. Finally, studies showed that increased awareness and empathy among egalitarians led to increased support for collective action on behalf of minoritized social groups and causes.

Outcomes tended to be slightly more focused on avoidant-based metrics, similar to prejudice-avoidant egalitarianism. Moreover, comprising primarily self-report studies, this research stream continues to be limited in its ability to draw conclusions on actual behaviors or behaviors that are influenced beyond self-reports.

#### 3.5.3. Moderators

Four types of moderators emerged from the studies in this domain: regulatory focus, group status, identity-blindness, and implicit bias. In exploration of how this belief differentially influences White vs. Black participants, researchers found that anti-egalitarianism among White participants and egalitarianism among Black participants predicted the tendency to rely on hypodescent (i.e., the categorization fallacy of ascribing the lower-status race to a multiracial person) when categorizing biracial targets. That is, White people who endorsed structural inequalities and Black people who opposed structural inequalities were more likely to categorize biracial targets as Black. Among Black participants, this effect was due to perceptions of shared fate, such that Black participants perceived biracial targets as likely to experience discrimination and felt a sense of similarity based on social group membership.

Regulatory focus differences in how participants approached the topic of intergroup equality impacted their physiological and behavioral speech outcomes. The regulatory focus theory suggests that distinct types of goals, those that are oriented toward the attainment of pleasure and those that are oriented toward the avoidance of harm, influence the achievement of goals. When positively oriented egalitarian goals were set, a challenge behavioral response was elicited, motivating the active pursuit of that goal. Alternatively, when obligatory goals were set, a threat response was initiated, leading to avoidance behaviors. Thus, beyond individual differences in beliefs about egalitarianism, the framing of social equality can have an impact on intrapersonal perception, physiology, and decisions to take social action.

Another identified moderator was identity-blind ideology. Identity-blindness led to positive outcomes when participants strongly endorsed the social hierarchy but led to negative outcomes when participants endorsed egalitarianism. People who support social hierarchical differences may be more tolerable or pleasant during intergroup settings if they ignore group membership (i.e., adopt identity-blind ideology), whereas egalitarians may be more harmful. In organizational contexts, color-blindness has been connected with more positive outcomes for people of color if and when organizational settings contain low levels of diversity [79]. Extending this work, beliefs about structural equality are an important determinant when predicting the effectiveness of certain types of interventions or policies to increase support for minoritized employees.

Finally, the interaction of egalitarianism and implicit bias appeared in this domain, which is part of a larger, emerging field of study on the phenomenon of aversive racism [80]. Aversive racism occurs when people who endorse egalitarian values harbor implicit bias or negative associations of an outgroup, influencing their intergroup behaviors. One study showed that egalitarian medical professionals who had greater levels of implicit bias spent less time assessing and diagnosing Black patients as compared to White patients, influencing downstream diagnostic accuracy and pain perceptions [76].

#### 3.5.4. Critical Summary

The most recent conceptualizations of egalitarianism have consisted of structural beliefs that disavow systems serving to uphold social inequalities. A majority of these beliefs have been measured using the Social Dominance Orientation scale and, due to the robust interest in SDO beliefs, reflect a negative bias that focuses on harmful outcomes of anti-egalitarianism. Despite this negative focus, research in this area has found that people who were low in social dominance orientation (egalitarians) were more attuned to the existence of social inequality, felt more empathy towards people who were targets of this inequality, were more motivated to strive towards social equality, and were more supportive of legislative and political action to change barriers preventing equality. Further, egalitarians attended to and valued different social cues from anti-egalitarians, showing a perceptual bias between these two groups.

Unlike previous conceptualizations of egalitarianism, low social dominance seems to operate as a belief that filters perception and intergroup beliefs, ultimately influencing behavior. Rather than reflect beliefs about individuals directly, like universal orientation or concern for others, low social dominance beliefs seem to operate in conjunction with other processes such as attention and empathy to guide egalitarian efforts. Indeed, this more recent set of contributions to the literature adds insight into the mechanisms through which egalitarianism is enacted or subverted, accounting for the influencing role of social structures and inequity beliefs.

Lacking in this conceptual framework, though, is how these attitudes, perceptions, and emotions contribute to the quality of individual-level intergroup interactions cross-sectionally and over time. Perceptions of disadvantages and empathetic emotions influence supportive attitudes and behaviors towards equality-driven structural change, but it is unknown how these processes influence interactions at the interpersonal level. Focusing solely on structure-level indicators of a person’s identity may lead to a reliance on stereotypes and increased implicit bias. For instance, people who interact with individuals based on preconceived knowledge derived from their social membership may be more likely to rely on stereotypes and engage in microaggressions, despite benevolent intentions [81]. Thus, knowledge about and concern for structural inequalities may not profoundly improve positive intergroup interactions on an individual level.

The present studies construed egalitarianism as an economic and politically oriented construct, primarily in relation to group-level attitudes and resource allocation behaviors. Systemic change undoubtedly relies on policy development and support. However, future research should investigate how awareness and support for social change influences other levels of analysis regarding intergroup relations [8].

## 4. Discussion

The purpose of this paper is to bring awareness to the lack of cohesion in defining and interpreting the concept of egalitarianism within intergroup paradigms. In recent years, attention has turned from prejudice reduction and toward understanding the development of positive intergroup interactions like allyship and collective action. Researchers have explored influences for these positive interactions through a variety of factors like motivational source, attitudes, cognitive bias, social contact, and values [82]. Despite this burgeoning interest and varied approach, research streams have developed independently from a centralized and unifying theoretical framework. To stimulate a reinvigorated focus on developing positive intergroup relations, we reviewed the existing literature on egalitarianism, a concept that may provide a fruitful starting point in establishing such a framework due to the influence values have on shaping behavior. Lacking a central model, we found considerable variations in conceptual definitions that could be categorized into five different forms of egalitarianism: prejudice avoidance, universal orientation, concern for others, positive expression, and low social dominance orientation. Within each of these categories, we found a diverse array of self-report measures, manipulation techniques, primes, and observational methods. To highlight the ways in which egalitarianism impacted intergroup relations, we assessed whether the observed effects were beneficial or harmful for intergroup relations and whether they motivated approach-oriented or avoidant-oriented reactions.

Most of the studies in this review characterized egalitarianism as prejudice avoidance, resulting in outcomes such as low prejudice, bias, stereotyping, and discrimination. These effects may set the stage for future positive intergroup relations, particularly where racism, sexism, and other forms of bias are deeply entrenched. However, without an additional measure of how non-prejudice or egalitarianism contributes to more proactive behaviors, this research stream only reveals factors that contribute to or predict low levels of prejudice and not necessarily intergroup flourishing.

Importantly, this conceptualization seems to diverge from at least one conceptualization held by people of color, that positive, proactive expressions from White people like smiling, being nice and helpful, seeking out friendly interactions, being authentic, and addressing prejudice in others were perceived as positive and preferred by people of color [25]. The majority of researchers’ avoidant-based conceptualizations differ from these indicators, conveying a discrepancy between research conclusions and lived expectations. This discrepancy underscores the importance of accurately and consistently developing a construct to guide psychological research that aligns with diverse expectations.

In a recent review, Urbioloa and colleagues highlight the effects of prejudice reduction models on cooling motivations to take collection action against social inequalities [83]. While evidence supports prejudice reduction can lead to more positive perceptions of outgroup members (i.e., harmony) and lower levels of discriminatory events, it can result in a backlash effect where more positive perceptions reduce the motivation and sense of urgency in combating institutional prejudice.

Another popular conceptualization, universal orientation seemed to be characterized by multiple processes that led to a mix of both positive and negative outcomes. Lacking an internal source, self-reported egalitarian beliefs and overt social norms tended to reveal positive, approach-oriented outcomes like volunteerism, positive outgroup attitudes, better work–life balance for women, and perceptions of similarity among diverse social group members. Similar to prejudice avoidance, many outcomes were also avoidant-based, such as reduced interracial anxiety and negative outgroup attitudes and low instances of discrimination. Manipulations of egalitarian self-concepts or egalitarian primes, however, led to some negative outcomes like aggressive, online behavior towards egalitarian-identifying women and stereotyping of egalitarian men. A correlation with the Protestant work ethic revealed that there may be a portion of universal orientation that is conflated with the belief in working hard to attain social equality. The Protestant work ethic has been positively associated with authoritarianism and negatively associated with humanistic values [84]. Additionally, aversive racism is shown to be a phenomenon among individuals with purported egalitarian goals who nevertheless succumb to implicit biased tendencies [80]. Without a central framework from which they measure and interpret the concept, researchers and participants may unintentionally be applying their assumptions of “egalitarianism” and responding to divergent signals. This collection of studies reveals the importance of clearly defining and measuring the network of parameters associated with a concept because of the sporadic narrative of associations that may occur otherwise.

Concern for others captured person-centered beliefs that contributed to positive, approach-oriented outcomes such as positive attitudes towards outgroup members, diverse online contact, and positive identity development among youth with minority social identities. However, with support from only four studies and relying exclusively on self-report methods, there is limited understanding as to where these beliefs stem from and how these attitudes may directly influence intergroup relations over time and across contexts. A recent report on national racial attitudes showed that White individuals’ endorsement for the principles of racial equality, support for the implementation of equalizing policies, and perceptions of closeness with racial outgroup members has steadily increased over the past several decades [85], yet since 2016, 65% of Americans agree that it has become more common for people to express racist sentiments [5]. Understanding underlying motivational sources that influence positive beliefs, such as concerns for others, and how these beliefs influence (or do not) behavior across diverse contexts can help elucidate the discrepancy between alleged social justice progress and reported racism being increasingly observed.

Positive expression consisted of a behavioral conceptualization whereby participants either indicated their intended or measured positive expressions toward outgroup members. Self-reported intent was related to warm, friendly behaviors and pro-diversity attitudes. Negative effects, like moral licensing and increased implicit bias, were seen among manipulated measures of egalitarianism. Lastly, a meta-analysis showed that clear, egalitarian social norms that promoted positive intergroup behavior predicted favorable outcomes for Black people, but the absence of these norms predicted an ingroup bias. Positive expression captures an essential function of egalitarianism that should not be overlooked due to its small presence in the literature. Unlike previous conceptualizations, positive expression was generated in attempt to understand socially privileged groups’ positive behavioral responses toward marginalized social groups (i.e., immigrants, Black people, etc.)

Whereas positive and engaging behaviors from privileged groups are preferred as expressions of non-prejudice [25], in some circumstances, positive behaviors can induce more harm than good, as displayed when teachers attempted to provide positive feedback to their students of color despite their need for opportunities to grow. That positive expressions and behaviors are desired in general does not negate the importance in determining motivational sources from which these expressions stem. For example, recent research has shown that racial minority participants who are suspicious of White people’s motives can reliably detect differences in positive expressions that stem from a desire to avoid prejudice versus authentic expressions of kindness [86]. When positive expressions were perceived as inauthentic, minority participants experienced greater cardiovascular threat and stress and a decrease in self-esteem [87]. Thus, positive expressions towards outgroup members are not a panacea for creating positive intergroup dynamics. Rather, authentic processes that motivate these expressions should be more deeply uncovered through multi-method studies and diverse measurements.

Lastly, low social dominance captured beliefs about the system-level impact on social relations. Although high social dominance beliefs are the central focus in this stream of research, results revealed that low social dominance beliefs were related to positive outcomes like perceptual focus, empathy, and support directed towards disadvantaged social groups. Demographic characteristics, like (dis)advantage status, race, and gender, were important in determining both the outcome and the nature of the outcome. For instance, although positive contact between groups improved attitudes, disadvantaged people who held positive attitudes towards advantaged people tended to focus on and believe less in social inequality, impacting their subsequent support for social change. Overall, it seems that for many groups, anti-hierarchy beliefs are related to positive attitudes toward disadvantaged groups and support for social change.

Although the lens of low social dominance presents a new frame of analysis to consider (e.g., systemic behaviors and attitudes), missing from this set of studies is the way in which these beliefs directly impact intergroup behaviors. For instance, do these beliefs foster motivation to engage or support at the interpersonal level? What cognitive or emotional mechanisms bidirectionally influence these beliefs? As a relatively new topic within the intergroup scope, this research stream provides considerable insight into the mechanisms that motivate intergroup beliefs and allows researchers to expand the study of egalitarianism to capture individual and system-level observations.

## 5. Conclusions and Future Directions

In the current state, an evaluation of egalitarianism within intergroup contexts has not yielded clear insight on whether or how it promotes or hinders intergroup flourishing. The ambiguous conclusions come not from a lack of interest, creativity, or scientific rigor, but rather, from a lack of organization in theoretical development. Indeed, over the past fifty years that psychologists have explored the nature of egalitarianism in diverse contexts, a central framework for the concept has not been developed.

As the positive psychological field is shifting towards diversifying content and amplifying its focus on positive social change, there stands an opportunity to centralize the intra- and interpersonal mechanisms that delineate egalitarian processes and contribute to positive intergroup development. Harnessing prior work in this field to scaffold a centralized model will provide insight into the distinction between authentic, internalized egalitarianism as a guiding value from externalized egalitarianism as a social obligation. By situating this model with the positive psychological lens, outcomes and correlates of interest can be measured in terms of growth, allophilia, flourishing, and positive social change, rather than harm, avoidance, or tolerance.

This also provides opportunities for cross-discipline collaboration, such as the sociological implications and predictors of egalitarian-driven dynamics. For instance, while the environment and social norms have been observed to influence individual behavior, e.g., [51], sociologists may be interested in how behaviors motivated from authentic egalitarianism influences or changes observers, social norms, and, ultimately, social institutions.

Moreover, a cohesive and expanded view of egalitarianism that is rooted in positive psychological processes can enhance the development of pragmatic interventions and trainings that can result in sustained social change for egalitarian societies [88,89,90,91]. An avenue rich with opportunities is that of allyship that spans contexts like everyday encounters (active bystanders and upstanders for marginalized groups), the workplace, and within personal networks. Allyship consists of behaviors that are often forecasted with much thought, reflection, and past emotional experiences [89]. Supplementary to the AMIGAS model [84], allyship inherently consists of collective action between individuals from privileged and disadvantaged groups. Studies of long-term, committed allies have found that they tend to have multiple experiences with both internal regulatory processes to govern bias reduction as well as positive, other-focused values that motivate continued expression that end up influencing future behaviors and decision-making [10]. Applying the lens of egalitarianism could explain and help develop these experiences into a two-pronged approach to building intergroup relationships.

## Figures and Tables

**Table 1 behavsci-14-00842-t001:** Prejudice-avoidance egalitarianism.

Author	Sample	Measure	Outcome	Moderator
Plant and Devine (1998) [9]	N = 1743 U.S. college students	Scale development: Internal Motivation to Respond without Prejudice (IMS), External Motivation to Respond without Prejudice (EMS)	Internal: (+ correlated) pro-Black attitudes, Attitudes Toward Blacks (ATB), Humanitarianism/Egalitarianism (H/E), (− correlated) modern racism, anti-Black attitudes, Right-Wing Authoritarianism (RWA), Protestant Work Ethic (PWE); guilt, shame, and self-criticism; External: (+) Modern Racism, Right-Wing Authoritarianism, social evaluation, (−) ATB, threat.	N/A
Moskowitz, et al. (1999) [14]	N = 242 German male students	Manipulation: Participants forced to respond stereotypically about women on a fixed-survey. Less stereotypical follow-up responses denoted egalitarianism	Lower subsequent stereotype scores were related to the inhibition of implicit stereotype activation.	N/A
Livingston and Drwecki (2007) [15]	N = 208 White American students	ATB [16], the Implicit Association Test (IAT) [17]	Low bias scores were associated with lower susceptibility to negative affect and more susceptibility to positive affect.	N/A
Mendes et al. (2007) [18]	N = 78 U.S. White adults	IAT [17]	Lower implicit racial bias scores were correlated with lower threat appraisals, less anxiety, and higher salutary neuroendocrine products when being interviewed by a Black research confederate	N/A
Amodio et al. (2008) [19]	N = 73 White U.S. students	IMS, EMS [9] ATB [16]	High IMS was related to positive explicit attitudes toward Black people but only high IMS/low EMS was related to better stereotype inhibition on implicit tasks compared to high IMS and high EMS; High IMS/low EMS showed better conflict monitoring activity.	EMS moderated the extent to which highly IMS successfully inhibited stereotype activation.
Castelli and Tomelleri (2008) [20]	N = 452 Italian students	IAT [17] self-reported perceptions of discriminatory norms	IAT responses were correlated with lower acceptability of discriminatory norms and quicker access to egalitarian words on a lexical decision task following a Black prime	The presence of others facilitated lower implicit bias and quicker egalitarian concept access compared to being alone
Johns et al. (2008) [21]	N = 164 White U.S. students	IMS, EMS [9], ring measure of social values (RSV) for egalitarian goal activation [22]	Participants with high IMS displayed more egalitarian giving in a points allocation task. In a subsequent task, high IMS predicted less stereotype activation on a lexical decision measure, mediated by egalitarian goal activation (RSV task).	IMS, Black prime
Plant and Devine (2009) [23]	N = 431 White U.S. students (across 3 studies)	IMS, EMS [9]	High EMS predicted amount of time spent decreasing detectable prejudice whereas high IMS and EMS predicted amount of time spent decreasing detectable and undetectable prejudice.	Feedback about possessing bias influenced high IMS, low EMS to reduce undetectable prejudice.
Wellman et al. (2009) [24]	N = 57 White American college students	ATB [16]	Confronting a racist joke	Witnessing a prior confrontation (exemplar); optimism
Winslow et al. (2011) [25]	N = 236 Black students	Open-ended prompt: “What are some things that you have personally seen or heard a white person do or say that made you think that person was not prejudiced?” and “In general, what can white people do to let you know they are not prejudiced, if they really are not?”	Smiling, greeting, or helping (24.5%), equal treatment (10.8%), confronting prejudice in others (10.8%), actively seek interactions (4.7%) and relationships (5.3%) with minorities; Avoid racist statements (17.6%), behave in authentic behaviors/just “be yourself” and do not overcompensate (69.1%), treat people equally (17.7%)	N/A
Legault and Green-Demers (2012) [26]	N = 377 Canadian students	Motivation to be Non-prejudiced scale (MNPS) [27]; IAT [17]	Self-determined prejudice regulators showed less modern racism, negative affect, interracial anxiety, implicit bias, and discrimination, and more positive affect	N/A
Schmader et al. (2012) [28]	N = 143 White American college students	IMS, EMS [9]	IMS (not EMS) predicted negative emotional reaction towards anti-diversity viewpoints and physiological threat reactivity towards anti-diversity viewpoints.	When a Black confederate was present and appeared angry, IMS predicted slightly more negative emotion.
Skorinko et al. (2015) [29]	N = 307 college students in America and Hong Kong	Manipulation: experimenter wore a t-shirt that read “People don’t discriminate, they learn it”.	Egalitarian prime exposure led to lower implicit and explicit bias toward homosexual targets in the collectivistic conditions	Cultural background: Collectivistic conditions led to positive outcomes whereas individualistic conditions had no effect.
Aranda and Montes-Berges (2016) [30]	N = 474 Spanish students	Manipulation: Text highlighting global gender inequality. Self-report: 30 -item scale of Prejudice, Egalitarian Commitment, Awareness of Inequality	In a decision-making task where participants assigned male and female targets to differing roles, female targets were assigned to more mid- and high-level positions.	Less stereotypical role-placement was observed more in stereotypically feminine companies; Women displayed less stereotypical role-placement
Gabarrot and Falomir-Pichastor (2017) [31]	N = 82 French college students	Manipulation: Information conveyed that majority of ingroup did not consider favoring ingroup against outgroup in terms of social welfare, housing, or education benefits, to be legitimate.	Higher identification with an ingroup predicted more prejudice toward the outgroup and negatively predicted stereotyping.	When high group similarity was primed with egalitarian norms, participants were more prejudiced.
Falomir-Pichastor et al. (2018) [2]	N = 506 Swiss college students	Manipulation: see Gabarrot and Falomir-Pichastor (2017) [31]	High prejudice justification predicted a preference for ingroup and high prejudice against the outgroup.	Within the egalitarian norm condition, ingroup preference and prejudice was higher among high prejudice justifiers if participants engaged in prior egalitarian behaviors than when they had not.
LaCosse and Plant (2020) [32]	Study 1: N = 99 Black undergraduate students Studies 2–6: Total N = 553 White undergraduate students and MTurk workers	Study 1: open-ended prompt “In general, when interacting with White people, what actions, behaviors, topics of conversation, etc., do they perform that lead you to feel respected?” Studies 2-6: self-reported intentions to show respect, avoid prejudice, focus on self/partner, IMS, EMS, memory recall (Study 5 only), observed engagement (Study 6 only)	Study 1: non-prejudiced and unbiased behaviors (incl active acknowledgement of social justice), rejecting stereotypes, genuine engagement. Study 2–6: IMS predicted intentions to show respect, focus on self and partner, avoid prejudice, and engage; better memory recall of partner; more engagement behaviors (EMS predicted prejudice concerns and self-focused intentions)	IMS and EMS
Szekeres et al. (2023) [33]	N = 1116 (across 3 studies) ranging from Black, Muslim, and Latinx Americans, and Romanian and Hungarian adults	IMS [9]	Participants with stronger egalitarian values were more likely to hypothetically confront bias than lower value counterparts. However, egalitarians were less likely to actually confront bias, compared to their predictions.	Participants higher in behavioral uncertainty were even less likely to confront.
Najdowski et al. (2024) [34]	N = 108 non-Black U.S. adults	IMS [9]	Higher internal motivations were related to reduced likelihood of promoting anti-Black social media post compared to egalitarian post.	N/A
Estevan-Reina et al. (2024) [35]	N = 1010 women (across 4 studies)	Manipulation of a man confronting sexism using egalitarian claims	Women found men who confronted sexism with egalitarian language, compared to paternalistic language, as better allies, felt more empowered, and perceived less of a power differential between themselves and the male ally.	N/A

**Table 2 behavsci-14-00842-t002:** Universal orientation egalitarianism.

Author	Sample	Measure	Outcome	Moderator
Beere et al. (1984) [36]	N = 367 adults	Scale development: sex-role Egalitarianism Scale	Women scored higher than men, psychology students scored higher than business students, students scored higher than police officers and senior citizens.	N/A
Gaertner et al. (1994) [37]	N = 1357 high school students	Self-reported perceptions of equal status, cooperative interdependence, association and interaction, and supportive norms	Egalitarian social norms were negatively correlated with emotional bias and negative attitudes toward racial outgroup members.	N/A
Gaertner et al. (1996) [38]	Study 3: N = 229 U.S. adults	Self-reported perceptions of equal status, cooperative interdependence, association and interaction, and supportive norms	Positive contact conditions were related to less social anxiety and lower sociability bias.	N/A
Phillips and Ziller (1997) [39]	N = 664	Scale development: Universal Orientation scale	(+) pro-Black attitudes, HE, PWE, (−) modern racism, anti-Black attitudes	N/A
Monteith and Walters (1998) [40]	N = 244 non-Black students	Self-report items that measured HE and PWE [41]	Among high prejudiced, HE led to moral obligation to regulate prejudice whereas PWE did not	Individual difference in perceptions of attainment of egalitarianism moderated prejudice regulation
Dall’Ara and Maass (1999) [42]	N = 120 Italian male students	Manipulation: female target described as occupying a nontraditional role who was not afraid to compete with a man and pursued equal work rights	Men were more likely to sexually harass (sent a pornographic image) and needed fewer persuasion attempts to sexually harass egalitarian targets compared to traditional targets.	Higher scores on measures of sexism, masculine identity, and likelihood to sexually harass
Dovidio and Gaertner (2004) [43]	Review	Combination of self-report and observational	Self-reported egalitarians engaged in fewer emergency and non-emergency helping behaviors, less support for affirmative action, greater perceptions of guilt among Black vs. White criminals, discriminatory hiring practices if they could justify their behavior on some aspect of the environment.	Environment ambiguity/justification
Chandler et al. (2009) [44]	N = 124 U.S. students	Universal Orientation Scale [39]	Higher scores predicted higher value, understanding, and career-based motivations to engage in volunteer behaviors.	N/A
Wyer (2010) [45]	N = 100 students	Manipulation: five-minute essay response to “All people are equal; therefore, they should be treated the same way”	Egalitarian prompt (compared to control) inhibited activation of avoidant words among high in prejudice	N/A
Rudman, Mescher and Moss-Racusin (2012) [46]	N = 518	Manipulation: male target described as supporting women’s rights, enjoying women’s studies, or having been involved in women’s issues research	Explicit and implicit favoring of egalitarian target (compared to sexist target) but was rated as more feminine, more gay, less masculine	Women showed more favorability of egalitarian target than men.
Lyness and Judiesch (2014) [47]	N = 40,921 individuals and 36 countries	Gender Inequality Index, Gender Egalitarian Practices, Gender Egalitarian values	Women and men report no difference in work–life balance in high gender egalitarian countries compared to low; Supervisors relied less on stereotypes when appraising women’s work-life conflict in high gender egalitarian countries compared to low.	N/A
Pahlke et al. (2014) [48]	N = 137 U.S. elementary students	Manipulation: five, 30 min gender pro-egalitarian lessons in identifying, analyzing, responding to gender stereotyping, biased judgements, unequal gender relationships; Self-report egalitarian attitudes (Preschool Occupation, Activity, and Trait-Attitude Measure)	Gender egalitarian lessons increased ability to identify sexism in the media and provide antisexist challenges in response to sexist remarks compared to control lessons.	N/A
Falomir-Pichastor et al. (2017) [49]	N = 521 adults (across 2 studies)	Manipulation: newspaper article about social equality stating “In sum, we are all equal and all groups should be equally treated” with statistics that 90% of Americans agree	Men reported more psychological differences between gay and straight men following egalitarian norms, negative attitudes toward gay people.	Promotion of biological similarities between groups; women did not show this pattern
Rosenblum et al. 2022 [50]	N = 1515 White and Black U.S. adults across 3 studies	Modern Racism Scale [51], IMS [9], Open-ended response: “Do you believe all people are equal and should have equality of opportunity?”	Black perceivers accurately detected White racial attitudes and motivations from egalitarian writing prompts. Humanizing language and support for equal opportunity were more indicative of lower prejudice and internal motivations.	N/A

**Table 3 behavsci-14-00842-t003:** Concern for others egalitarianism.

Author	Sample	Measure	Outcome	Moderator
Katz and Hass (1988) [41]	N = 202 U.S. White students	Scale development: Humanitarianism-Egalitarianism (HE) scale	(+) pro-Black attitudes, (−) anti-Black attitudes	N/A
Monteith and Spicer (2000) [57]	N = 496 White and 275 Black students	Open-ended essay response to “I have generally positive/negative feelings toward Black/White people because…”	White participants: HE (+) with more positive essay themes and negatively correlated with modern racism; Black participants: No correlation between value orientation and essay themes	Racial Identity
Schwab and Greitemeyer (2015) [58]	N = 357 adults from 18 countries	HE [41], self-reported outgroup attitudes	HE (+) with number of outgroup friends on social media and positive outgroup attitudes	N/A
Villegas-Gold and Tran (2018) [59]	N = 383 multiracial U.S. adults	Self-report egalitarian socialization scale	(+) with integrated identification, well-being, and self-esteem	Higher racial ambiguity was associated with more egalitarian socialization and more positive outcomes.

**Table 4 behavsci-14-00842-t004:** Positive expression egalitarianism.

Author	Sample	Measure	Outcome	Moderator
Aberson and Ettlin (2004) [60]	N = 31 studies (meta-analysis)	Coded articles for egalitarian contexts and norms	Within clear egalitarian contexts, Black targets were favored; within ambiguous contexts, AA targets received more negative ratings and behaviors.	N/A
Plant et al. (2010) [61]	N = 230 non-Black U.S. students	IMS [9]	High IMS more likely than low IMS to report focusing on approach goals/strategies for upcoming interracial interaction and more likely to use these strategies during a real interracial interaction (resulting in more pleasant interaction).	Among high internally motivated P’s, high external motivation (EMS) reduced positive interaction quality.
Harber et al. (2010) [62]	N = 108 White U.S. student teachers	Manipulation: Results of a “social issues survey” that conveyed either a pro- or anti-minority slant; Participants had to provide five examples of famous Black individuals from easy or difficult categories (government vs. math).	When egalitarian self-images were threatened, teachers gave more positive essay feedback, recommended less time for developing writing skills, and supplied more positive comments to students they thought were Black compared to students they thought were White, regardless of student writing ability.	N/A
Mann and Kawakami (2012) [63]	N = 225 students	Manipulation: instructed to “try to have positive evaluations when a Black target appeared on screen” which would be monitored by a progress bar that showed they made progress toward or away from this goal.	Progressing on egalitarian goals (+) with sitting further away from the Black research confederate, closer to the White research confederate, and greater implicit bias.	N/A
Van Bergen et al. (2017) [64]	N = 22 Turkish, Moroccan, or Dutch youth	Interviews of youth on attitudes toward outgroup members, experiences with disparate treatment, and parents’ responses to these events	Youth with positive outgroup attitudes reported that their parents had multicultural friend groups, taught about fallacies of stereotypes, and encouraged positive intergroup interactions (parent–child similarity).	N/A

**Table 5 behavsci-14-00842-t005:** Low social dominance orientation egalitarianism.

Author	Sample	Measure	Outcome	Moderator
Saguy et al. (2009) (only study 2) [65]	N = 175 Israeli citizens	Questionnaire about contact, attention to illegitimate aspects of inequality (‘‘To what extent would you consider the inequality between the groups as just?’’), outgroup fairness, and support for social change.	Positive contact between advantaged and disadvantaged groups increased disadvantaged members’ positive attitudes toward the outgroup, which in turn increased perceptions of the outgroup as fair, decreased attention to inequality, and decreased support for social change.	N/A
Does et al. (2012) [66]	N = 37 Dutch college students	Behavior manipulation task: participants asked to give oral presentation via webcam about equality in terms of ideals or obligations and how they could attain the ideal or obligation of social equality.	Speech tasks given from an ideal perspective elicited greater relative challenge than the obligation perspective. Behaviorally, obligation-framed speeches were spoken more slowly than ideal-framed speeches, indicative of self-monitoring.	Regulatory focus
Ho et al. (2015) [67]	N = 3107 American adults recruited from various online platforms	Scale development: subdomain (SDO egalitarianism) of social dominance orientation	(+) system legitimacy beliefs, political conservatism, support for unequal intergroup distribution of resources, opposition to hierarchy attenuating social policies; (−) concern for harm and fairness.	Higher status groups (men and White people) had higher levels of social dominance beliefs.
Ho et al. (2017) [68]	N = 2731 American adults recruited from various online platforms	Social Dominance Orientation (SDO) [67]	Both Black and White participants categorized biracial targets as more Black than White (hypodescent) but this was moderated by SDO only among White participants. Black participants use of hypodescent was negatively correlated with SDO and mediated by perceptions of discrimination against biracial people and sense of shared fate.	White hypodescent derivative of high SDO whereas Black hypodescent is derivative of perceived discrimination and shared fate.
Kteily et al. (2017) [69]	N = 1221 American adults recruited from various online platforms (3 of 5 studies included)	SDO [67]	SDO (−) perceptions of power differences between groups, which predicted support for egalitarian social policy. Incentivization to report honestly had no effect on the relationship between SDO and perceptions of lessened inequality, indicating differences in actual perceptions of inequality not just motivations.	N/A
Yogeeswaran et al. (2017) [70]	N =4599 New Zealand adults	SDO [71,72]	Low SDO: colorblindness negatively predicted outgroup warmth	SDO moderates the effects of colorblindness.
Kende et al. (2018) [73]	N = 459 studies including 660 samples in 36 countries	Schwartz Value Surveys (SVS): “How important is equality (equal opportunity for all) as a guiding principle in your life?”; GNI (index of inequality)	The negative correlation between intergroup contact and prejudice was stronger in countries with higher cultural egalitarianism, above and beyond equal status situational factors. This link was weaker in countries with stronger hierarchy-enforcing culture. Thus, cultural and situational equality produce most optimal outcomes for intergroup contact and prejudice reduction.	N/A
Lucas and Kteily (2018) [74]	N = 2340	SDO [67]	Increased empathy with disadvantaged targets compared to advantaged targets. When the target was disadvantaged, SDO (−) empathy; when the target was advantaged, SDO (+) empathy. High SDO (−) perceptions of harm against disadvantaged targets and subsequently predicted low levels of empathy. Among advantaged targets, SDO (+) perceived harm and high levels of empathy. (−) between SDO and empathy among disadvantaged targets was stronger than the (+) between SDO and empathy among advantaged targets.	Degree to which high SDO individuals felt empathy, perceived harm, or opposed detrimental policies depended on whether or not the target was in an advantaged or disadvantaged group.
Coksan and Cingoz-Ulu (2022) [75]	N = 146 Kurdish and Turkish participants	Self-report on ingroup norms	When group members perceive ingroup norms as egalitarian, social identity has no effect on ingroup bias and resource allocation.	N/A
Do Bu et al. (2023) [76]	N = 617 White, Portuguese medical trainees (across 5 studies)	SDO [60]	Medical professionals who hold egalitarian values still spent less time assessing and diagnosing Black patients compared to White patients, resulting in lower diagnostic accuracy, reduced pain perception accuracy, and increase opioid prescription behaviors.	Egalitarians with higher implicit bias engaged in this “aversive racism” behavior.
Hoyt et al. (2023) [77]	N = 255 undergraduate American students	SDO [60]	Anti-egalitarians, but not egalitarians, who scored higher in color evasiveness (or colorblindness) indicated less support for Black student activism, reduced engagement in social justice, lower satisfaction with student leaders, lower perceived effectiveness of demonstrations, and higher beliefs in activism as causing intergroup conflict.	N/A

## Data Availability

Data are contained within the article.

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
