# Peer review of "Exploring Egalitarianism: A Conceptual and Methodological Review of Egalitarianism and Impacts on Positive Intergroup Relations"

_behavsci, 2024, doi:10.3390/bs14090842_

Round 1
Reviewer 1 Report
Comments and Suggestions for Authors
This article is excellent. At first, I was confused about what was happening in the literature as I worked through each type of egalitarianism in the Results section. While I normally am not favorable to lengthy discussion sections that repeat the review of the literature, in this case the Discussion helped me tie together the authors’ point about the ambiguity (or I might say inconsistency) of the findings. In other words, I did not understand this article reached the Discussion.
The one point at which I felt my confusion still justified in the Results section was when nonintuitive or contradictory results were highlighted. At these points I could have used more details about what was found and perhaps possible explanations. The extended discussion of moderating factors when they appeared was a nice feature of this article.
I have no major methodological issues with this article. The authors are clear about their search strategy and provide enough detail about each article to help the reader determine whether the reader should read the original article.
As I read this article, I initially wondered to myself whether there was a need to turn sociological to look at structural effects and cultural influences as a source of ambiguity. The article occasionally took steps in this direction when raising questions of culture and the environment, but simply stating that culture and environment may be predictors gives us little insight into how and why. Linking the psychological and sociological considerations may be a promising line of research that could have been mentioned in the Conclusion starting at line 911.
Reviewer 2 Report
Comments and Suggestions for Authors
The article, ‘A Methodological Review and Research Agenda on Egalitarianism to Foster Positive Intergroup Relations,’ presents a thoughtful and innovative exploration of egalitarianism and its role in fostering positive intergroup relations. The authors have set themselves an ambitious task by undertaking a systematic review of a wide range of literature to develop a comprehensive framework that goes beyond the traditional focus on prejudice reduction. This shift towards a positive psychology perspective is a valuable contribution to the field, as it offers new avenues for understanding and promoting harmonious intergroup interactions. The manuscript is well structured and persuasive, with a clear delineation of the various aspects of egalitarianism, and offers valuable insights to the ongoing discourse in social psychology and intergroup relations.
Nevertheless, several aspects of the manuscript could be further refined to enhance its clarity, coherence, and impact.
Firstly, while the concept of egalitarianism is a central tenet of the manuscript, its definition and application are somewhat inconsistent throughout the text. At times, the term "egalitarianism" is defined as the absence of prejudice (163), while in other instances, it is used to describe a broader range of positive cross-group behaviors (747-754). A clear and consistent definition of the term "egalitarianism" at the beginning of the manuscript would help to avoid confusion and ensure that readers have a solid understanding of the concept as it is used throughout the work.
Furthermore, the manuscript would benefit from a reduction in redundancy, particularly in the discussion of internal and external motivations for prejudice reduction. These concepts are revisited on numerous occasions, often with similar explanations. The consolidation of these sections would enhance the manuscript's readability, enabling the authors to direct their attention to novel insights rather than reiterating established ideas. A more succinct delineation of these pivotal concepts would also render the manuscript more accessible to a broader readership. A more concise presentation of these key concepts would also facilitate broader accessibility of the manuscript.
The methodological approach adopted by the authors is sound; however, how the methods are discussed and critiqued is clearly in need of improvement. Although the manuscript provides a comprehensive overview of the methods employed in the studies examined, a more detailed analysis of the rationale behind the selection of specific methods and their influence on the outcomes would considerably enhance the value of the discussion.
In this context, the heavy reliance on self-reporting should be reflected by either discussing the limitations of these measures or suggesting alternative methods that could provide more nuanced insights into egalitarian behaviors.
In conclusion, the manuscript represents a significant contribution to the field of prejudice research by presenting a positive psychology approach to understanding egalitarianism and intergroup relations. With some refinements, such as clarifying key concepts, reducing redundancies, improving methodological rigor, and expanding the discussion of practical implications, the manuscript has the potential to become an even more impactful and influential scholarly work.
